# Antilisterial and Antimicrobial Effect of *Salvia officinalis* Essential Oil in Beef Sous-Vide Meat during Storage

**DOI:** 10.3390/foods12112201

**Published:** 2023-05-30

**Authors:** Robert Gál, Natália Čmiková, Aneta Prokopová, Miroslava Kačániová

**Affiliations:** 1Department of Food Technology, Faculty of Technology, Tomas Bata University in Zlín, Vavrečkova 275, 760 01 Zlín, Czech Republic; gal@ft.utb.cz; 2Institute of Horticulture, Faculty of Horticulture and Landscape Engineering, Slovak University of Agriculture, Tr. A. Hlinku 2, 94976 Nitra, Slovakia; 3Department of Polymer Engineering, Faculty of Technology, Tomas Bata University in Zlín, Vavrečkova 275, 760 01 Zlín, Czech Republic; 4Department of Bioenergetics, Food Analysis and Microbiology, Institute of Food Technology and Nutrition, Rzeszow University, Cwiklinskiej 1, 35-601 Rzeszow, Poland

**Keywords:** antimicrobial effect, *Listeria monocytogenes*, beef tenderloin (*m. psoas major*), sous-vide, *Salvia officinalis* essential oil

## Abstract

If food is contaminated with pathogens such as *Listeria monocytogenes*, improper cooking during sous-vide preparation can lead to foodborne illnesses. In this study, it was found that *L. monocytogenes* were inactivated with both heat and the essential oil of *Salvia officinalis* (sage EO) in beef tenderloin of the *musculus psoas major* that had undergone sous-vide processing. To determine whether the enhancement of the efficacy of heat treatment is prospective, *L. monocytogenes* and sage EO were mixed. Groups with *L. monocytogenes* alone and sage essential oil combined with *L. monocytogenes* and test groups without EO were established. The samples were vacuum-packed, inoculated with *L. monocytogenes*, and then cooked sous-vide for the predetermined duration at 50, 55, 60, or 65 °C. In both groups with sous-vide beef tenderloin, the total bacterial count, the coliforms bacterial count, and the amount of *L. monocytogenes* were assessed on days 0, 3, 6, 9, and 12. Over these days, the amounts of *L. monocytogenes*, coliform bacteria, and overall bacteria increased. The identification of bacterial strains in various days and categories was performed by MALDI-TOF mass spectrometry. The test group that was exposed to a temperature of 50 °C for 5 min had a higher overall bacterial count for each day that was assessed. *Pseudomonas fragi* and *L. monocytogenes* were the most isolated organisms from the test group and the treated group. To ensure the safety for the consumption of sous-vide beef tenderloin, it was found that the addition of natural antimicrobials could produce effective outcomes.

## 1. Introduction

In contrast to conventional cooking techniques, sous-vide involves cooking food in vacuum-sealed containers at precisely controlled temperatures, resulting in a better flavor, texture, and nutritional value as well as a longer shelf life [1]. The evaluation of microbial safety is critical in this cooking technique, so it is essential to understand how this treatment affects microorganisms in order to evaluate the safety of products [2]. The authors of the study [3] found that pathogens present in foods prepared by the sous-vide method at the time of ingestion came from raw ingredients as they had not been destroyed by cooking [3]. The range of temperatures between 30 and 50 °C, where bacterial growth and reproduction are first inhibited, is ideal for the development of most pathogenic bacteria. The temperature of the food during preparation should not be below 54.4 °C to ensure the inactivation of food pathogens, such as *Salmonella* species, *Listeria monocytogenes*, and *Escherichia coli* pathogenic strains [4].

To date, there is not enough evidence to support the antilisterial properties of sage essential oil (EO) in foods. A small number of authors have described how sage affects both Gram-positive and Gram-negative bacteria [5]. *Listeria monocytogenes* can become less resistant to heat treatment and sage chemicals either through a synergistic combination or simply as a result of the cumulative effects of the individual components on microorganisms [6]. The essential oil of *Salvia officinalis* disrupted the cell membrane and altered its permeability, causing the release of several cytoplasmic components, such as macromolecular compounds, ATP, and DNA. Sage essential oil has a broad antibacterial effect that is due not only to a special route mechanism but also to a number of activities on the cell surface and in the cytoplasm. To fully understand the antibacterial mechanism of essential sage oil, further research is needed [7]. Natural food preservation methods have recently attracted a lot of attention from both consumers and food technologists [8]. According to research by Korczak et al. [9], the inclusion of sage significantly reduced the development of unpleasant flavors and odors in precooked meat products while they were stored at 4 °C. Madsen et al. [10] claim that sage can be used successfully to prevent the emergence of a warmed-over flavor and thus improve the sensory quality of heat-treated meat products. Gas chromatography/mass spectrometry (GC/MS) and gas chromatography (GC-FID) were used to identify the major compounds in *S. officinalis* EO, including α-thujone (24.6%), camphor (20.6%), 1,8-cineole (12.1%), and α-humulene (5.8%) [11].

The impact of sage EO during meat preservation is a subject that has not yet been thoroughly investigated. Sage could be added to beef patties, according to Zhang et al. [12], without having a negative impact on the sensory qualities of the burger. Natural dried sage powder, even in high concentrations, has been shown to be effective in preserving the sensory quality of cooked beef burgers, according to Mizi et al. [13]. Turkey meatballs with sage extract exhibited superior sensory quality attributes compared to test samples, according to Karpinska-Tymoszczyk [14]. According to data, sage EO may help limit the development of *L. monocytogenes*. Sage may be used as a natural preservative, but to successfully limit microbial growth it must be combined with other substances [15].

Fresh beef vacuum packaging has been shown to be effective in extending shelf life and preserving the sensory characteristics of the product for a long period of time. By limiting oxidation and the development of aerobic microorganisms during cooling, the vacuum increases the shelf life of the meat. The vacuum packaging method has been used more frequently in the institutional market for the distribution of whole pieces of beef [16]. 

The purpose of this experiment was to study the behavior of *Listeria monocytogenes* that had been inoculated into beef meat with the addition of sage essential oil and to observe the effect of vacuum packing, various heat treatments, and 12 days of storage time. To simulate the storage at a temperature commonly used by consumers in the refrigerator, a temperature of 6 °C was used. The total number of microorganisms and their identification were studied.

## 2. Materials and Methods

### 2.1. Sample Preparation

In this experiment, beef meat samples from the thigh (*m. psoas major*) were used. According to the information on the label produced in the Czech Republic, the meat sample was obtained from Charolais breading that was purchased from an authorized retailer. The meat samples were delivered to the microbiological facility in a clean refrigerator under hygienic conditions, where they were kept at 6 °C until the analyses were performed. Within 120 min, the samples were moved from the approved store to the laboratory. The meat was diced and samples weighing 5 g were treated with 1% SOEO solutions (Hanus, Nitra, Slovakia), dissolved in sunflower oil, and vacuum-packaged using a vacuum packer (Concept, Choceň, Czech Republic). Good-quality sunflower oil was purchased from an authorized dealer. A total of 480 different beef samples were examined. The samples studied were treated in the following manner: 

BM—fresh beef meat was vacuum-packaged in polyethylene bags, stored anaerobically at 6 °C, and treated at 50–65 °C for 5–25 min.

BMLMEO—fresh beef meat treated with *L. monocytogenes* and 1% EO sage was vacuum-packaged in polyethylene bags, stored anaerobically at 6 °C, and treated at 50–65 °C for 5–25 min. 

The control samples were prepared from uncooked raw meat on day zero. Essential oils were added to the samples and maceration was performed for 24 h. The samples were placed in the CASO SV1000 sous-vide device. *L. monocytogenes* CCM 4699 was prepared at 1.5 × 10^8^ CFU and added to the sample at a volume of 100 µL.

### 2.2. Samples Cultivation

Microbiological tests were performed at 6 °C on days 0, 1, 3, 6, and 12. Samples of five grams were diluted in an Erlenmeyer beaker with 45 mL of a 0.1% sterile saline solution. The samples were homogenized for 30 min in the GFL 3031 shaking incubator of Burgwedel, Germany. The microbial communities were examined: Violet Red Bile Lactose Agar (VRBL, Oxoid, Basingstoke, UK) was used for coliform bacteria culture and incubated at 37 °C for 24 to 48 h. Total viable counts (TVCs) were grown on Plate Count Agar (PCA, Oxoid, Basingstoke, UK), which was incubated at 30 °C for 48 to 72. Then, the total viable counts in this medium were calculated. A 0.1 mL sample was used to inoculate Oxford Agar with an Oxford supplement (Oxoid, Basingstoke, UK) for *L. monocytogenes count*. Incubation took place at 37 °C for 24 h.

### 2.3. Identification of Microorganisms by MALDI-TOF MS 

Using the MALDI-TOF (Matrix-Assisted Laser Desorption/Ionization Time of Flight) MS Bio-typer (Bruker, Daltonics, Bremen, Germany) and reference libraries, microorganisms isolated from beef meat samples were identified.

As an organic substance, a stock solution was created. The standard solution contained 2.5% trifluoroacetic acid, 47.5% water, and 50% acetonitrile. Amounts of 500 mL of pure 100% acetonitrile, 475 mL of purified water, and 25 mL of pure 100% trifluoroacetic acid were combined to create 1 mL of stock solution. The organic solvent was made and combined with “HCCA matrix portioned” in a 250 L Eppendorf flask. All of the substances used to prepare the matrix were purchased from Lambda Life (Bratislava, Slovakia).

The samples were prepared according to the previous instructions [17]. Eight colonies per Petri dish were briefly examined. In an Eppendorf flask, biological material was transferred from a Petri plate along with 300 μL of distilled water, then it was mixed and 900 μL of ethanol was added. The mixture was then centrifuged for two minutes at 10,000× *g* (ROTOFIX 32A, Ites, Vranov, Slovakia). The precipitate was removed from the Eppendorf tube after the supernatant was removed and left to dry at room temperature 20 °C. The particle was then treated with 30 L of 70% formic acid and 30 μL of acetonitrile. The mixture was then centrifuged for 2 min at 10,000× *g*. A MALDI plate was coated with 1 μL of liquid, which was then followed by the addition of 1 μL of a MALDI matrix solution. The samples were dried before being processed for microorganism identification on a MALDI-TOF mass spectrometer (Bruker, Daltonics, Bremen, Germany). The LT MALDI-TOF microflex mass spectrometer (Bruker Daltonics, Bremen, Germany) was used to create mass spectra automatically and was set to work in a linear positive mode with a mass range of 2000–20,000 Da. The device was calibrated using the Bruker bacterial test standard. The results of the mass spectra were examined using MALDI Bio-typer 3.0 software (Bremer, Germany-based Bruker Dal-tonics). The following were the identification criteria: scores between 2.300 and 3.000 denoted a highly probable species identification; scores between 2.000 and 2.299 secured a genus identification with a probable species identification; scores between 1.700 and 1.999 denoted a probable genus identification; and a score less than 1700 was considered an unreliable identification.

### 2.4. Statistical Analysis

Triplicates of each test and analysis were performed. The mean and standard deviation (SD) of the microbial numbers were calculated using Microsoft Excel. Using Prism 8.0.1 (GraphPad Software, San Diego, CA, USA), one-way analysis of variance (ANOVA) was carried out prior to a Tukey’s test with a significance level of 0.05. Data analysis was carried out using SAS^®®^ software version 8. 

## 3. Results

### 3.1. Total Count of Bacteria

In our investigation, the total numbers of bacteria in the control group and the group treated with sage EO and *L. monocytogenes* were assessed on day 0. The number of total bacteria counts ranged in the control group from 2.20 ± 0.07 (50 °C, 20 min) to 2.50 ± 0.13 log CFU/g (50 °C, 5 min) and in the treated group from 1.91 ± 0.07 (50 °C, 20 min) to 2.24 ± 0.07 log CFU/g (50 °C, 5 min) (Table 1). 

There was no count of microorganisms in sous-vide beef meat that had been heated to higher temperatures. The total number of bacteria decreased on day 0 in counts from the time used for treatment. The number of coliform bacteria on day 0 was zero. The number of *L. monocytogenes* decreased in the treated group with time used at a temperature of 50 °C (Figure 1).

Table 2 shows the impact of sage EO for each temperature treatment on day 3 in sous-vide beef samples. Average counts obtained in samples with or without sage EO over time and in accordance with heat treatment are shown in this table. The highest number in the test group was found in samples treated at 50 °C for 5 min. The number of coliforms bacteria on day 3 was zero. The number of *L. monocytogenes* in the treated groups ranged from 3.30 ± 0.10 log CFU/g (50 °C, 20 min) till 3.57 ± 0.12 log CFU/g (50 °C, 5 min) (Figure 2).

The antimicrobial effect of the treatment of sage EO, temperature, and time on day 6 are shown in Table 3. The total count of bacteria in the control group ranged from 2.30 ± 0.08 log CFU/g (55 °C, 20 min) to 2.87 ± 0.09 log CFU/g (50 °C, 5 min) and in the group with treatment of sage EO and *L. monocytogenes* from 2.09 ± 0.03 log CFU/g (55 °C, 5 min) to 2.70 ± 0.06 log CFU/g (50 °C, 5 min). The number of coliform bacteria on day 6 was zero. The number of *L. monocytogenes* decreased in the treated group with the time used at a temperature of 50 °C (Figure 3).

The total count of bacteria (Table 4) ranged in the control groups from 2.21 ± 0.12 log CFU/g (60 °C, 20 min) to 3.40 ± 0.04 log CFU/g (50 °C, 5 min) and in the treated groups from 1.17 ± 0.04 log CFU/g (60 °C, 20 min) to 3.22 ± 0.13 log CFU/g (50 °C, 5 min). The total count of bacteria for groups of sous-vide beef meat treated at a 65 °C temperature was zero. The total number of bacteria decreased on day 9 in counts from the time used for treatment. The number of coliform bacteria (Table 5) on day 9 ranged from 2.18 ± 0.12 log CFU/g (50 °C, 20 min) to 2.45 ± 0.04 log CFU/g (50 °C, 5 min) in the control groups and ranged from 1.28 ± 0.09 log CFU/g (50 °C, 20 min) to 1.85 ± 0.06 log CFU/g (50 °C, 5 min) in the treated groups. The number of *L. monocytogenes* decreased in the treated group by the time used at a temperature of 50 °C (Figure 4).

The antimicrobial effects of sage EO, temperature, and time on day 12 are shown in Table 6. The total count of bacteria in the control group ranged from 1.90 ± 0.08 log CFU/g (65 °C, 10 min) to 3.90 ± 0.10 log CFU/g (50 °C, 5 min) and in the group with treatment of sage EO and *L. monocytogenes* from 2.05 ± 0.03 log CFU/g (65 °C, 10 min) to 3.64 ± 0.08 log CFU/g (50 °C, 5 min). The number of coliform bacteria (Table 7) on day 12 in the control group ranged from 2.20 ± 0.11 log CFU/g (55 °C, 5 min) to 3.26 ± 0.05 log CFU/g (50 °C, 5 min) and in the group with treatment of sage EO and *L. monocytogenes* from 2.41 ± 0.05 log CFU/g (55 °C, 5 min) to 2.82 ± 0.04 log CFU/g (50 °C, 5 min). The number of *L. monocytogenes* in the treated groups ranged from 4.95 ± 0.16 log CFU/g (50 °C, 20 min) to 5.41 ± 0.04 log CFU/g (50 °C, 5 min) (Figure 5).

### 3.2. Isolated Species of Bacteria

A total of 381 isolates were identified from sous-vide beef meat from the control and treated groups of samples. A total of 10 families, 14 genera, and 25 species were isolated from the control group of samples (Figure 6). The most isolated species in this study were *Pseudomonas fragi* (21.53%), *Hafnia alvei* (10%), and *Pantotea agglomerans* (8.9%) followed by *Kocuria salcida* (7%). A total of 8 families, 13 genera, and 21 species were isolated from the treated group of sous-vide beef meat (Figure 7). The most isolated species was *L. monocytogenes* (28%), which was added to this group. The other most isolated species of bacteria from the treated group were *P. fragi* (10%), *Lysinibacillus xylanitaticus* (6%), *H. alvei* (5%), and *Pseudomonas graminis* (5%).

## 4. Discussion

Antimicrobials can be added to food to ensure the microbial safety of ready-to-eat meals. The main role of antimicrobials is to prevent or eliminate pathogens and spoilage microorganisms. Due to concerns about synthetic preservatives, customers have recently preferred foods made with natural antimicrobials. Organic compounds or essential oils of plants, for example, have been used to prevent food from spoiling [18]. They have also been used to help ensure food safety, which was the focus of this research. Our results show the influence of the essential oil of *S. officinalis* at different temperatures and times. Higher temperatures and longer times have proven to be the most effective.

Sous-vide cooking involves the vacuum packing of raw or partially cooked food, which is then cooked at specific times and temperatures, chilled, and kept below 6 °C. This technique maintains the visual appeal and nutritional content of the food [19,20,21,22]. Vacuum sealing in low oxygen permeability pouches suppresses some of the odors associated with oxidation and prevents the evaporation of moisture and volatiles during cooking. Sarcoplasmic proteins are collected at temperatures up to 65 °C, increasing their softness, and nutritional losses decrease because nutrients are not sucked out by boiling water [19,21,23]. Sage is known for its anti-inflammatory effects [24] and antibacterial action against various bacteria, such as *Enterococcus faecalis* ATCC29212, *Klebsiella pneumoniae* ATCC700603, *Salmonella Paratyphi* A NCTC13, and *Staphylococcus aureus* ATCC29213) and spoilage microorganisms (*Proteus mirabilis*, *Photobacterium damselae*, *Vibrio vulnificus*, *Enterococcus faecalis*, *Pseudomonas luteola*, and *Serratia liquefaciens*) [25]. Several authors have documented this in various food matrices. The cause is reported to be 1,8-cineole, camphor, and thujone [6,26,27,28,29]. Sage EO in foods has an antilisterial effect; however, this effect is not well understood. According to a small number of writers, there is some evidence that sage exerts bactericidal and bacteriostatic effects on Gram-positive and Gram-negative bacteria [5]. Through a synergistic impact or simply through the additional effects that each component has on microorganisms, combining sage chemicals with heat treatment may reduce the resistance of *L. monocytogenes* [6].

The results of the study show the behavior of *Listeria monocytogenes* inoculation in beef with the addition of sage essential oil and vacuum packaging, over a period of 12 days and under different temperature settings. Our findings demonstrate the development of *L. monocytogenes*, coliform bacteria, and total bacteria from day 0 to day 12. In a separate study, long cooking times at low temperatures were used to mimic the conditions that arise in retail food service when processed foods are prepared sous-vide. According to Tangwatcharin et al. [30], different sous-vide temperatures for inoculated restructured goat steak had an impact on the D-values of *L. monocytogenes*. Its Z-value was 8.20 °C, and its D-value decreased as the temperature increased. Non-inoculated restructured goat sirloin was cooked using six D-values at 60, 65, and 70 °C to ensure the safety of the sous-vide product. The number of microorganisms in all samples was reduced, and pathogens were not found after cooking using various sous-vide techniques.

The optimal duration and temperature for cooking salmon sous-vide have been estimated to eliminate *L. monocytogenes*, and oregano oil and citric acid may aid by reducing the bacteria’s ability to withstand heat. The study’s findings are important for ensuring the safety of food and may help processing centers lower the risk of *L. monocytogenes* during thermal treatment. Our findings will also shed light on potential uses for heat treatment that could improve the results [31].

Low temperatures had a bacteriostatic impact on *L. monocytogenes*, according to Chan and Wiedmann [32]. This conclusion is consistent with this study’s findings, which indicate that control samples stored at 2 °C significantly decreased from day 0 to day 28 by 1.23 log (on average). In our research, a consistent decrease in *L. monocytogenes* counts was observed during storage at 4 °C. An exponential increase was observed up to day 21.

The growth curves of *L. monocytogenes* inoculated in beef at storage temperatures ranging from 5 to 25 °C were observed in a study by Lee et al. [33]. At 5 °C, *L. monocytogenes* was discovered to thrive. Calculating the lag and stationary phase was impossible because of the very slow rate of development at this temperature. According to Farber and Peterkin [34], the *L. monocytogenes* lag phase in vacuum-packed roast beef held at 3 °C lasted 59 h, illustrating the effect of stress heat treatment on the organism and resulting in a log lag phase. The lack of growth of *L. monocytogenes* during storage prevented us from observing a lag period in our research at 2 °C. The multiplication of *L. monocytogenes* in a vacuum-packed mortadella kept at 4 °C and 8 °C was reported [35].

After heat treatment (grilling, microwave heating, or conventional cooking), Yilmaz et al. [36] noted a decrease in the number of mesophilic microorganisms. After 15 days of storage, the mesophilic bacterial counts in the study samples with additives were comparable to those found after thermal processing. The count of mesophilic bacteria did not alter significantly during the storage of vacuum-packed cooked turkey breast rolls, according to Smith and Alvarez [37]. In our study, different families, genera, and species were discovered in the sous-vide beef meat treatment group and the control test. The *P. fragi* species was the most isolated. The activity of various strains of this species under various packaging conditions may explain the seemingly contradictory findings. Consequently, more research is necessary to understand the prevalence and function of *Pseudomonas fragi* as meat spoilers [38]. The main spoiler of chilled beef kept in aerobic storage is thought to be *Pseudomonas fragi* [39]. Due to its fast development ability and the creation of strong odors associated with rotting, it quickly results in the meat being unacceptable for consumption [40]. It was also frequently discovered to be present in vacuum-packed beef [39,41,42,43], indicating the existence of several strains, likely with various genetic repertoires. The bacterium that was most frequently seen was *H. alvei*. The study by Sokołowicz et al. [44] discovered the same outcomes with isolated species.

## 5. Conclusions

The most effective combinations of sous-vide cooking time and temperature for beef meat to inactivate *L. monocytogenes* are determined by this study, and it is suggested that sage essential oil may be able to reduce the ability of *L. monocytogenes* to withstand heat in beef tenderloin treated with sous-vide technique. The results of this study are important for ensuring the safety of food and may help processing facilities lower the risk of *L. monocytogenes* during heat treatment. Our research will also shed light on potential uses that could improve the effects of thermal treatment when combined with proper times and sage essential oils.

## Figures and Tables

**Figure 1 foods-12-02201-f001:**
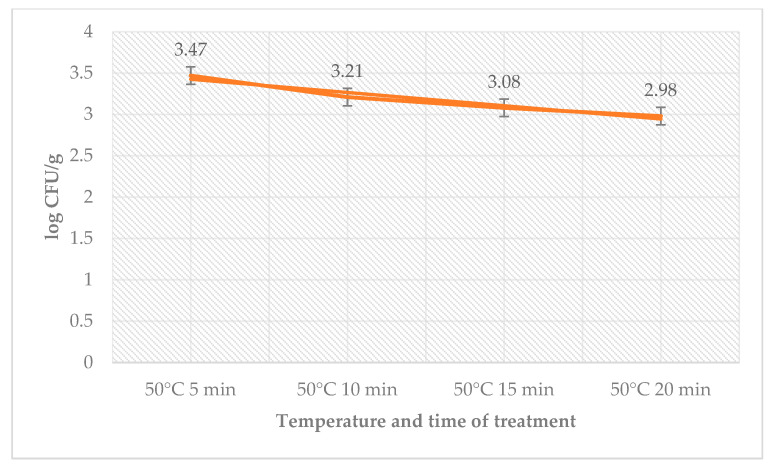
Number of *L. monocytogenes* (log CFU/g) on day 0 in the sage EO treatment group.

**Figure 2 foods-12-02201-f002:**
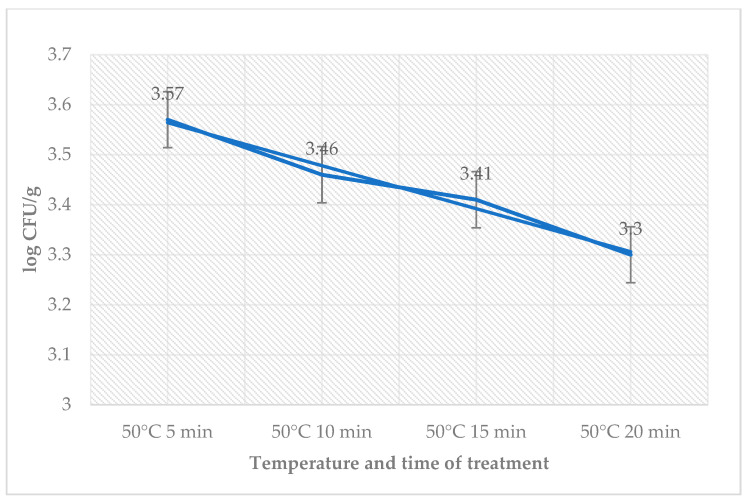
Number of *L. monocytogenes* (log CFU/g) on day 3 in the group treated with sage EO.

**Figure 3 foods-12-02201-f003:**
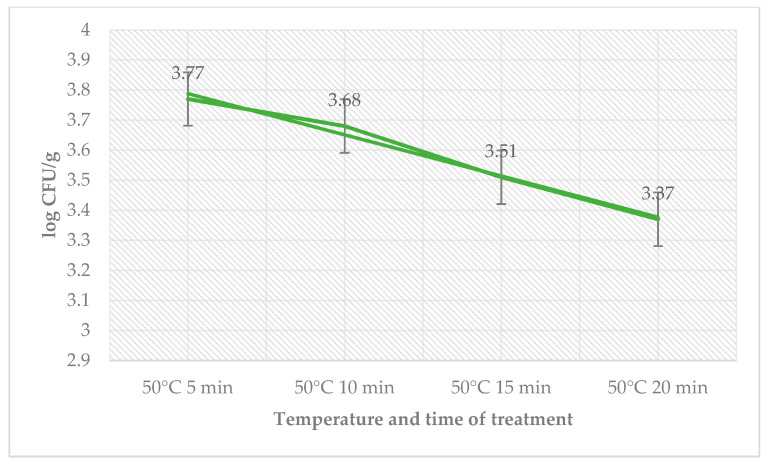
Number of *L. monocytogenes* (log CFU/g) on day 6 in the group treated with sage EO.

**Figure 4 foods-12-02201-f004:**
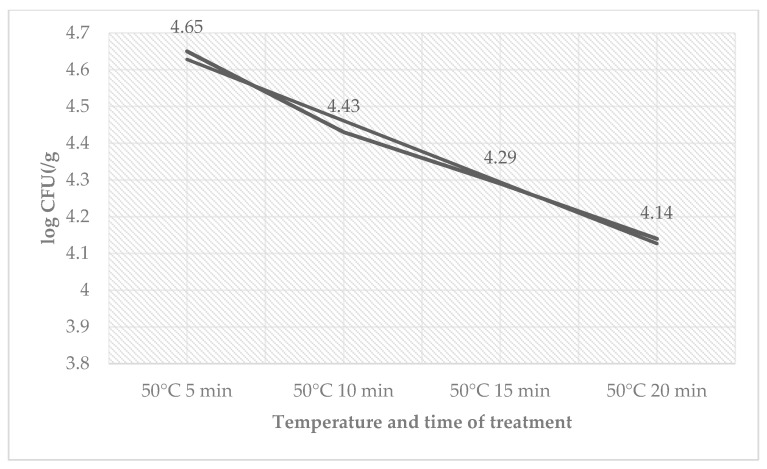
Number of *L. monocytogenes* (log CFU/g) on day 9 in the group treated with sage EO.

**Figure 5 foods-12-02201-f005:**
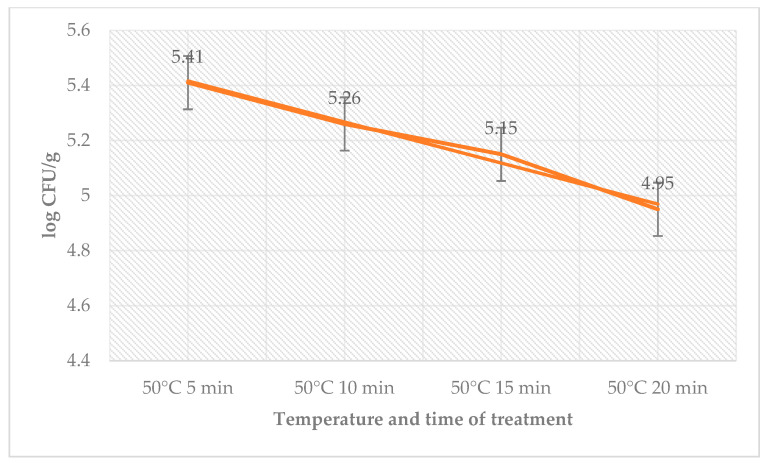
Number of *L. monocytogenes* (log CFU/g) on day 12 in the group treated with sage EO.

**Figure 6 foods-12-02201-f006:**
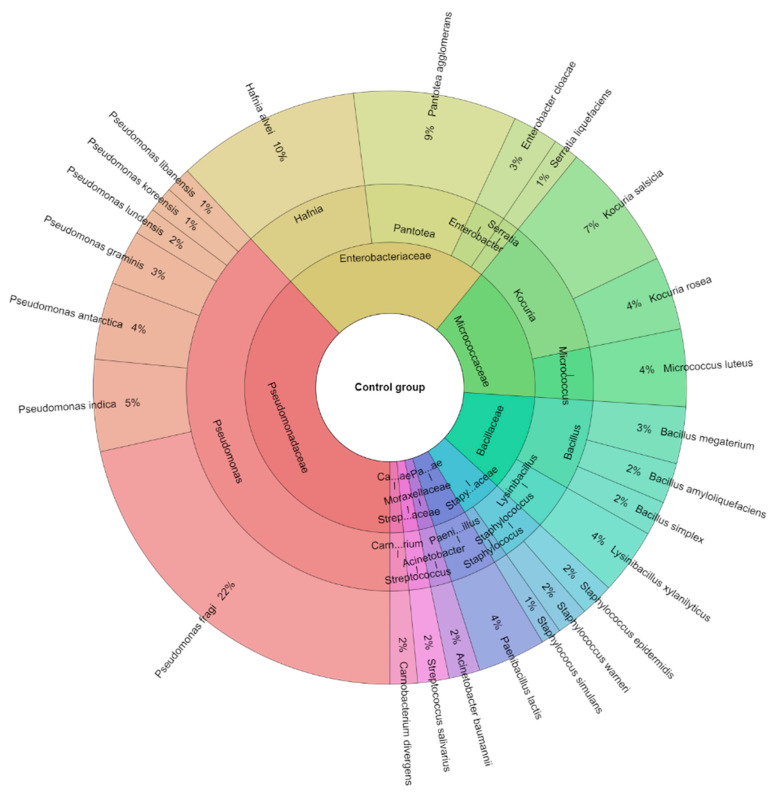
Krona diagram of isolated species of bacteria from the control group.

**Figure 7 foods-12-02201-f007:**
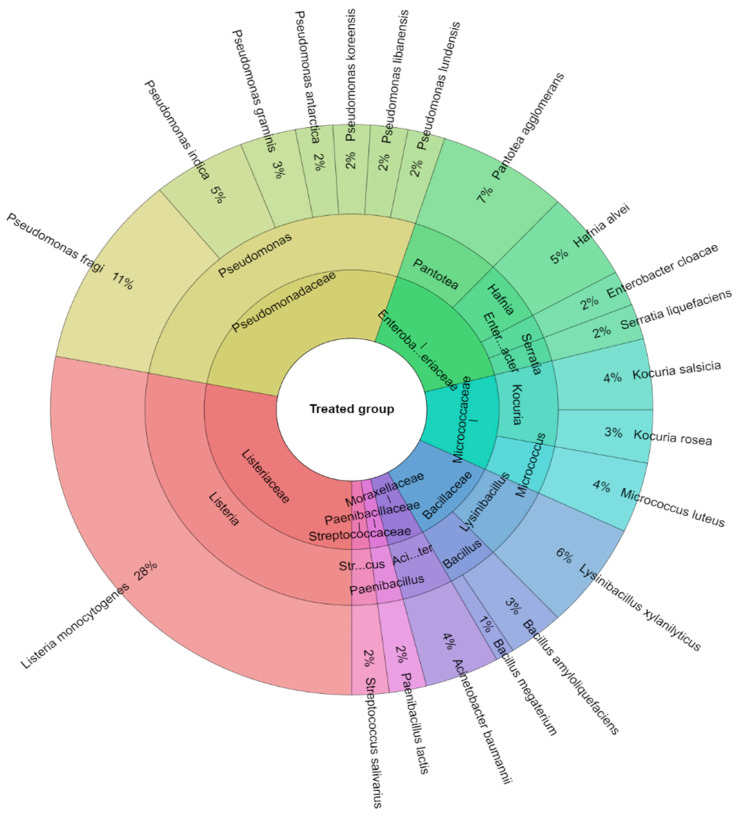
Krona diagram of isolated species of bacteria from the treated group.

**Table 1 foods-12-02201-t001:** The total count of bacteria in the control group and the group treated with sage EO and *L. monocytogenes* (log CFU/g) in 0 days.

Treatment	Temperature (°C)	Time (min)	Average	SD	*p* Value
BM	50	5	2.50	0.13	3.699 × 10^−2^
BMLMEO	50	5	2.24	0.07	
BM	50	10	2.42	0.06	5.546 × 10^−3^
BMLMEO	50	10	2.13	0.07	
BM	50	15	2.28	0.06	6.336 × 10^−3^
BMLMEO	50	15	2.03	0.06	
BM	50	20	2.20	0.07	6.139 × 10^−3^
BMLMEO	50	20	1.91	0.07	

BM: fresh beef meat was vacuum-packaged in polyethylene bags, stored anaerobically at 6 °C, and treated at 50–65 °C for 5–25 min; BMLMEO: fresh beef meat treated with *L. monocytogenes* and 1% of sage EO was vacuum-packaged in polyethylene bags, stored anaerobically at 6 °C, and treated at 50–65 °C for 5–25 min.

**Table 2 foods-12-02201-t002:** Total count of bacteria in the control group and group treated with sage EO and *L. monocytogenes* (log CFU/g) on day 3.

Treatment	Temperature (°C)	Time (min)	Average	SD	*p* Value
BM	50	5	2.63	0.06	8.301 × 10^−2^ *
BMLMEO	50	5	2.52	0.06	
BM	50	10	2.56	0.08	9.114 × 10^−2^ *
BMLMEO	50	10	2.45	0.03	
BM	50	15	2.47	0.05	1.890 × 10^−1^ *
BMLMEO	50	15	2.40	0.06	
BM	50	20	2.30	0.20	3.730 × 10^−1^ *
BMLMEO	50	20	2.44	0.12	
BM	55	5	2.36	0.02	3.290 × 10^−2^
BMLMEO	55	5	2.31	0.02	
BM	55	10	2.30	0.10	2.894 × 10^−1^ *
BMLMEO	55	10	2.23	0.08	
BM	55	15	2.25	0.07	1.721 × 10^−1^ *
BMLMEO	55	15	2.18	0.02	
BM	55	20	2.17	0.05	1.432 × 10^−1^ *
BMLMEO	55	20	2.09	0.05	

BM: fresh beef meat was vacuum-packaged in polyethylene bags, stored anaerobically at 6 °C and treated at 50–65 °C for 5–25 min; BMLMEO: fresh beef meat treated with *L. monocytogenes* and 1% of sage EO was vacuum-packaged in polyethylene bags, stored anaerobically at 6 °C and treated at 50–65 °C for 5–25 min. * The data are not statistically significant at the 95% significance level.

**Table 3 foods-12-02201-t003:** Total count of bacteria in the control group and group treated with sage EO and *L. monocytogenes* (log CFU/g) on day 6.

Treatment	Temperature (°C)	Time (min)	Average	SD	*p* Value
BM	50	5	2.87	0.09	4.144 × 10^−2^
BMLMEO	50	5	2.70	0.06	
BM	50	10	2.76	0.08	3.139 × 10^−2^
BMLMEO	50	10	2.59	0.06	
BM	50	15	2.67	0.09	7.019 × 10^−2^ *
BMLMEO	50	15	2.51	0.06	
BM	50	20	2.60	0.10	1.830 × 10^−1^ *
BMLMEO	50	20	2.44	0.08	
BM	55	5	2.43	0.06	3.411 × 10^−1^ *
BMLMEO	55	5	2.39	0.04	
BM	55	10	2.42	0.08	5.202 × 10^−2^ *
BMLMEO	55	10	2.24	0.08	
BM	55	15	2.37	0.05	6.755 × 10^−3^
BMLMEO	55	15	2.19	0.03	
BM	55	20	2.30	0.08	1.424 × 10^−2^
BMLMEO	55	20	2.09	0.03	

BM: fresh beef meat was vacuum-packaged in polyethylene bags, stored anaerobically at 6 °C, and treated at 50–65 °C for 5–25 min; BMLMEO: fresh beef meat treated with *L. monocytogenes* and 1% of sage EO was vacuum-packaged in polyethylene bags, stored anaerobically at 6 °C, and treated at 50–65 °C for 5–25 min. * The data are not statistically significant at the 95% significance level.

**Table 4 foods-12-02201-t004:** Total count of bacteria in the control group and group treated with sage EO and *L. monocytogenes* (log CFU/g) on day 9.

Treatment	Temperature (°C)	Time (min)	Average	SD	*p* Value
BM	50	5	3.40	0.04	1.194 × 10^−1^ *
BMLMEO	50	5	3.22	0.13	
BM	50	10	3.31	0.08	2.117 × 10^−3^
BMLMEO	50	10	2.89	0.07	
BM	50	15	3.26	0.08	3.525 × 10^−4^
BMLMEO	50	15	2.70	0.03	
BM	50	20	3.08	0.08	7.764 × 10^−4^
BMLMEO	50	20	2.58	0.05	
BM	55	5	3.05	0.06	6.324 × 10^−4^
BMLMEO	55	5	2.49	0.08	
BM	55	10	2.87	0.11	8.11 × 10^−4^
BMLMEO	55	10	2.20	0.07	
BM	55	15	2.80	0.06	5.917 × 10^−2^
BMLMEO	55	15	2.14	0.04	
BM	55	20	2.75	0.06	4.779 × 10^−2^
BMLMEO	55	20	1.95	0.06	
BM	60	5	2.38	0.07	3.222 × 10^−4^
BMLMEO	60	5	1.66	0.08	
BM	60	10	2.307	0.02	1.546 × 10^−2^
BMLMEO	60	10	1.59	0.05	
BM	60	15	2.23	0.06	2.175 × 10^−2^
BMLMEO	60	15	1.37	0.05	
BM	60	20	2.21	0.12	1.350 × 10^−4^
BMLMEO	60	20	1.17	0.04	

BM: fresh beef meat was vacuum-packaged in polyethylene bags, stored anaerobically at 6 °C, and treated at 50–65 °C for 5–25 min; BMLMEO: fresh beef meat treated with *L. monocytogenes* and 1% of sage EO was vacuum-packaged in polyethylene bags, stored anaerobically at 6 °C, and treated at 50–65 °C for 5–25 min. * The data are not statistically significant at the 95% significance level.

**Table 5 foods-12-02201-t005:** Coliform bacteria in the control group and the group treated with sage EO and *L. monocytogenes* (log CFU/g) on day 9.

Treatment	Temperature (°C)	Time (min)	Average	SD	*p* Value
BM	50	5	2.45	0.05	2.065 × 10^−4^
BMLMEO	50	5	1.85	0.06	
BM	50	10	2.35	0.04	6.221 × 10^−2^
BMLMEO	50	10	1.71	0.06	
BM	50	15	2.27	0.04	1.257 × 10^−2^
BMLMEO	50	15	1.52	0.04	
BM	50	20	2.18	0.04	5.653 × 10^−2^
BMLMEO	50	20	1.28	0.09	

BM: fresh beef meat was vacuum-packaged in polyethylene bags, stored anaerobically at 6 °C, and treated at 50–65 °C for 5–25 min; BMLMEO: fresh beef meat treated with *L. monocytogenes* and 1% of sage EO was vacuum-packaged in polyethylene bags, stored anaerobically at 6 °C, and treated at 50–65 °C for 5–25 min.

**Table 6 foods-12-02201-t006:** Total count of bacteria in the control group and group treated with sage EO and *L. monocytogenes* (log CFU/g) on day 12.

Treatment	Temperature (°C)	Time (min)	Average	SD	*p* Value
BM	50	5	3.90	0.10	2.685 × 10^−2^
BMLMEO	50	5	3.64	0.08	
BM	50	10	3.76	0.08	1.797 × 10^−1^ *
BMLMEO	50	10	3.58	0.17	
BM	50	15	3.64	0.07	4.083 × 10^−2^
BMLMEO	50	15	3.40	0.12	
BM	50	20	3.56	0.08	4.058 × 10^−2^
BMLMEO	50	20	3.35	0.09	
BM	55	5	3.47	0.07	6.631 × 10^−3^
BMLMEO	55	5	3.16	0.07	
BM	55	10	3.24	0.07	1.572 × 10^−2^
BMLMEO	55	10	3.04	0.05	
BM	55	15	3.14	0.06	1.110 × 10^−1^ *
BMLMEO	55	15	2.87	0.09	
BM	55	20	3.03	0.13	7.664 × 10^−2^ *
BMLMEO	55	20	2.64	0.11	
BM	60	5	2.71	0.07	1.133 × 10^−1^ *
BMLMEO	60	5	2.43	0.09	
BM	60	10	2.64	0.08	2.063 × 10^−1^ *
BMLMEO	60	10	2.40	0.10	
BM	60	15	2.45	0.08	4.512 × 10^−1^ *
BMLMEO	60	15	2.35	0.06	
BM	60	20	2.34	0.06	6.905 × 10^−2^ *
BMLMEO	60	20	2.17	0.03	
BM	65	5	2.20	0.10	1.325 × 10^−1^ *
BMLMEO	65	5	2.09	0.03	
BM	65	10	1.90	0.10	3.407 × 10^−1^ *
BMLMEO	65	10	2.05	0.08	

BM: fresh beef meat was vacuum-packaged in polyethylene bags, stored anaerobically at 6 °C, and treated at 50–65 °C for 5–25 min; BMLMEO: fresh beef meat treated with *L. monocytogenes* and 1% of sage EO was vacuum-packaged in polyethylene bags, stored anaerobically at 6 °C, and treated at 50–65 °C for 5–25 min. * The data are not statistically significant at the 95% significance level.

**Table 7 foods-12-02201-t007:** Coliform bacteria in the control group and group treated with EO and *L. monocytogenes* (log CFU/g) on day 12.

Treatment	Temperature (°C)	Time (min)	Average	SD	*p* Value
BM	50	5	3.26	0.05	1.229 × 10^−2^
BMLMEO	50	5	2.82	0.04	
BM	50	10	3.13	0.04	2.381 × 10^−2^
BMLMEO	50	10	2.77	0.04	
BM	50	15	3.00	0.10	8.283 × 10^−2^ *
BMLMEO	50	15	2.66	0.05	
BM	50	20	2.81	0.07	5.750 × 10^−2^ *
BMLMEO	50	20	2.54	0.04	
BM	55	5	2.20	0.11	2.397 × 10^−1^ *
BMLMEO	55	5	2.41	0.05	

BM: fresh beef meat was vacuum-packaged in polyethylene bags, stored anaerobically at 6 °C, and treated at 50–65 °C for 5–25 min; BMLMEO: fresh beef meat treated with *L. monocytogenes* and 1% of sage EO was vacuum-packaged in polyethylene bags, stored anaerobically at 6 °C, and treated at 50–65 °C for 5–25 min. * The data are not statistically significant at the 95% significance level.

## Data Availability

The data presented in this study are available on request from the corresponding authors.

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
