# Peer review of "Antilisterial and Antimicrobial Effect of Salvia officinalis Essential Oil in Beef Sous-Vide Meat during Storage"

_foods, 2023, doi:10.3390/foods12112201_

Round 1

Reviewer 1 Report

The manuscript entitled 'Antimicrobial effect of Salvia officinalis essential oil against Listeria monocytogenes in beef sous-vide meat during storage' provides meaningful insights in furnishing antilisterial activity of phytochemicals.

1. Lines 39-41: cite 'several studies', instead of single study.

2. It has been given in the discussion section the components of sage EOs. It would be better to narrate it in the introduction.

3. Line 290 mentions the activity of sage EOs against pathogens: mention the pathogens studied.

4. It is stated that the mechanism of action of sage EO against several pathogens except listeriae were understood. It would be fine to describe them in the discussion section.

5. Explain the antilisterial study procedure in the methodology section with suitable citation.

6. Use appropriate statistics in the representation. Use error bars in Fig. 4

7. Lines 296-297: Whether the sous-vide and application of sage EOs in this study were additive or synergistic. Narrate the interaction.

Author Response

Reviewer #1

The manuscript entitled 'Antimicrobial effect of Salvia officinalis essential oil against Listeria monocytogenes in beef sous-vide meat during storage' provides meaningful insights in furnishing antilisterial activity of phytochemicals.

Dear reviewer we are thank you for valuable comments. 

Point 1: Lines 39-41: cite 'several studies', instead of single study.

Response: it was corrected.

Point 2. It has been given in the discussion section the components of sage EOs. It would be better to narrate it in the introduction.

Response: It was moved to Introduction.

Point 3. Line 290 mentions the activity of sage EOs against pathogens: mention the pathogens studied.

Response: it was added.

Point 4. It is stated that the mechanism of action of sage EO against several pathogens except listeriae were understood. It would be fine to describe them in the discussion section.

Response: it was added to introduction.

Point 5. Explain the antilisterial study procedure in the methodology section with suitable citation.

Response: The methodology is not from different study.

Point 6. Use appropriate statistics in the representation. Use error bars in Fig. 4

Response: It was corrected.

Point 7. Lines 296-297: Whether the sous-vide and application of sage EOs in this study were additive or synergistic. Narrate the interaction.

Response: It was redescribed.

Reviewer 2 Report

Manuscript title: Antimicrobial effect of Salvia officinalis essential oil against Listeria monocytogenes in beef sous-vide meat during storage

Manuscript ID: foods-2343750

Authors: Robert Gál et al.

The study authored by Robert Gál et al. described the effect of sage oil on the Listeria monocytogenes during sous-vide cooking of beef. The authors collected beef and artificially inoculated them with L. monocytogenes. These spiked samples were treated with sage oil and were cooked by sous-vide method. The results were interesting as the authors could identify the combination of sage oil, time and temperature of cooking that effectively inhibited growth of L. monocytogenes.

With rise of consumer awareness and escalating avulsions to chemical preservatives, there is an increasing trend in employing natural antimicrobial substances to enhance the shelf life and product safety. In this context the study is important and meaningful. Moreover, the results of the study may be easily commercialized after refinement and may possibly be patented even.

However, there are certain weaknesses (listed below) in the manuscript that need to addressed by the authors.

General:

Manuscript requires substantial English editing to address clarity, grammatical errors and general improvement in readability.

E.g. Lines: 15-18, 45-46, 62-63, 73-75, 90, 350-352, and many others.

Abstract

1.      There is mismatch in between Title and Abstract.

2.      For example, the scientific name of sage (Salvia officinalis) was never mentioned.

3.      In the abstract, authors introduced non-Listeria bacteria which was not there in the Title.

4.      The authors concluded non-specifically on natural antibicrobials, while there research was on Salvia officinalis and Listeria monocytogenes (as per title).

Introduction

1.      Introduction section requires improvement.

2.      Authors need to elaborate previous studies to justify their choice of sage oil for antimicrobial effects, though they described the organoleptic aspects of sage oil.

Methods

1.      Line 89: Why 480 samples?

2.      Line 96-97: Which essential oils? How was the maceration performed? At what temperature?

3.      Did the authors check the samples to be free of Listeria monocytogenes on arrival, and before inoculations?

4.      Lines 102-103: Why were the microbiological tests carried out at 4 degree Celsius? This seems to be unusual.

5.      Lines 102-111: This section is written in non standard manner and needs revision.

6.      Did they follow any previously reported literature? If yes, please cite them. Else provide sufficient details for the experiments to be reproducible in other laboratories.

7.      How did the authors count Listeria monocytogenes in samples?

Results

1.      Lines 156-158: This experiment was not described in the methods section?

2.      Line 173: Not clear.

3.      Tables and figures being more in number, appear repetitive. Authors should, combine them to enhance readability.

4.      Lines: 261-270: Why did the authors isolate so many bacteria? That was not their study aim.

5.      Figure 6 and 7 Not needed.

Discussion

This section also requires revision especially providing possible explanation for the results obtained and also potential implications of the findings.

Author Response

Reviewer #2

Manuscript title: Antimicrobial effect of Salvia officinalis essential oil against Listeria monocytogenes in beef sous-vide meat during storage

Manuscript ID: foods-2343750

Authors: Robert Gál et al.

The study authored by Robert Gál et al. described the effect of sage oil on the Listeria monocytogenes during sous-vide cooking of beef. The authors collected beef and artificially inoculated them with L. monocytogenes. These spiked samples were treated with sage oil and were cooked by sous-vide method. The results were interesting as the authors could identify the combination of sage oil, time and temperature of cooking that effectively inhibited growth of L. monocytogenes.

With rise of consumer awareness and escalating avulsions to chemical preservatives, there is an increasing trend in employing natural antimicrobial substances to enhance the shelf life and product safety. In this context the study is important and meaningful. Moreover, the results of the study may be easily commercialized after refinement and may possibly be patented even.

However, there are certain weaknesses (listed below) in the manuscript that need to addressed by the authors.

Dear reviewer we are thank you for valuable comments. 

General:

Manuscript requires substantial English editing to address clarity, grammatical errors and general improvement in readability.

Point 1: E.g. Lines: 15-18, 45-46, 62-63, 73-75, 90, 350-352, and many others.

Response: It was redescribed.

Abstract

Point 2:   There is mismatch in between Title and Abstract.

Response: It was changed.

Point 3:   For example, the scientific name of sage (Salvia officinalis) was never mentioned.

Response: It was added.

Point 4:   In the abstract, authors introduced non-Listeria bacteria which was not there in the Title.

Response: Title was changed.

Point 5:  The authors concluded non-specifically on natural antimicrobials, while there research was on Salvia officinalis and Listeria monocytogenes (as per title).

Response: It was added.

Introduction

Point 6: Introduction section requires improvement.

Response: It was changed.

Point 7: Authors need to elaborate previous studies to justify their choice of sage oil for antimicrobial effects, though they described the organoleptic aspects of sage oil.

Response: it was changed.

Methods

Point 8: Line 89: Why 480 samples?

Response: It is 48 samples in one group, control group, group with essential oil and 5 different days, in three repetitions, together 480 samples.

Point 9:   Line 96-97: Which essential oils? How was the maceration performed? At what temperature?

Response: BM is without essential oil and BMLMEO is with sage essential oil. It is described. Samples were homogenized by the hand for one minute.

Point 10: Did the authors check the samples to be free of Listeria monocytogenes on arrival, and before inoculations?

Response: Yes, of course it was control samples in the experiment which were without Listeria monocytogenes.

Point 11: Lines 102-103: Why were the microbiological tests carried out at 4 degree Celsius? This seems to be unusual.

Response: It was mistake it is 6 °C.

Point 12:  Lines 102-111: This section is written in non standard manner and needs revision.

Response: It was rewritten.

Point 13: Did they follow any previously reported literature? If yes, please cite them. Else provide sufficient details for the experiments to be reproducible in other laboratories.

Response: Int the material and methods are all information for reproducible in other laboratories.

Point 14: How did the authors count Listeria monocytogenes in samples?

Response: L. monocytogenes were inoculated on Oxford agar with supplement and with Maldi- Tof MS identified.

Results

Point 15: Lines 156-158: This experiment was not described in the methods section?

Response: It was moved to Introduction.

Point 16: Line 173: Not clear.

Response: It was redescribed.

Point 17:      Tables and figures being more in number, appear repetitive. Authors should, combine them to enhance readability.

Response: In the tables is number of coliform bacteria and total count of bacteria, in the figures is number of L. monocytogenes on Oxford agar.

Point 18:      Lines: 261-270: Why did the authors isolate so many bacteria? That was not their study aim.

Response: It was corrected.

Point 19:      Figure 6 and 7 Not needed.

Response: The figure 6 and 7 is identified bacteria from study. Aim was corrected.

Discussion

This section also requires revision especially providing possible explanation for the results obtained and also potential implications of the findings.

Response: Some parts were redescribed.

Reviewer 3 Report

In the present manuscript the authors studied the effect of Salvia officinalis essential oil against Listeria monocytogenes in beef sous-vide meat during refrigerated storage. The authors performed different microbiological determinations (total bacterial count, coliforms count, and L. monocytogenes count) and identification of colonies by MALDI-TOF MS to study the evolution of bacterial populations during storage. In my opinion, this manuscript could be a valuable contribution for readers, considering the growing attention on the use of natural additives and new food cooking practices. Despite these positive aspects, in my opinion the manuscript needs a thorough review. Specific comments are given below.

L24-25 MALDI-TOF mass spectrometry

L43-46 perhaps you would like to mean that to avoid the incidence of the listed pathogens it is not recommended to drop below 54.4 degrees during cooking? Please rephrase this sentence because it is not clear

L47 Use “to date” instead “however”

L67-71 I suggest moving this part immediately after the description of antimicrobial activity of sage essential oil

L75 stored for 12 days

L76 Was the storage temperature of samples 6 °C? In the rest of the manuscript you always indicate a temperature of 4°C

L86 Please indicate the measure of samples also in cm

L91-99 Was the storage temperature of samples 4°C?

L102 Remove at 4°C

L113-146 I suggest merging the part on MALDI-TOF identification in one chapter

L119-125 Maybe you can remove the part about matrix preparation

L156-158 I find this information out of context. Please remove this part

L160-161 This is a consideration and should not be reported in the results section

TABLES 1-7 AND FIGURES 1-5. These tables and figures are structured identically and report the values of the different microbiological determinations at each sampling point. Providing a table and a figure with microbiological results for each sampling day is not very meaningful, because everything becomes very complicated and difficult to read. The authors should include all data in a maximum of one/two tables and one/two figures, so the reader can immediately visualize at a quick glance the evolution of the various microbiological parameters during the storage.

Author Response

Reviewer #3

In the present manuscript the authors studied the effect of Salvia officinalis essential oil against Listeria monocytogenes in beef sous-vide meat during refrigerated storage. The authors performed different microbiological determinations (total bacterial count, coliforms count, and L. monocytogenes count) and identification of colonies by MALDI-TOF MS to study the evolution of bacterial populations during storage. In my opinion, this manuscript could be a valuable contribution for readers, considering the growing attention on the use of natural additives and new food cooking practices. Despite these positive aspects, in my opinion the manuscript needs a thorough review. Specific comments are given below.

Dear reviewer we are thank you for valuable comments. 

Point 1: L24-25 MALDI-TOF mass spectrometry

Response: It was corrected.

Point 2: L43-46 perhaps you would like to mean that to avoid the incidence of the listed pathogens it is not recommended to drop below 54.4 degrees during cooking? Please rephrase this sentence because it is not clear

Response: It was changed.

Point 3: L47 Use “to date” instead “however”

Response: it was changed.

Point 4: L67-71 I suggest moving this part immediately after the description of antimicrobial activity of sage essential oil

Response: It was changed.

Point 5: L75 stored for 12 days

Response: It was added.

Point 6: L76 Was the storage temperature of samples 6 °C? In the rest of the manuscript you always indicate a temperature of 4°C

Response: It was corrected.

Point 7: L86 Please indicate the measure of samples also in cm

Response: It was corrected. It refers to weight in g.

Point 8: L91-99 Was the storage temperature of samples 4°C?

Response: It was corrected for 6 °C.

Point 9: L102 Remove at 4°C

Response: it was changed.

Point 10: L113-146 I suggest merging the part on MALDI-TOF identification in one chapter

Response: it was changed.

Point 11: L119-125 Maybe you can remove the part about matrix preparation

Response: It is necessary.

Point 12: L156-158 I find this information out of context. Please remove this part

Response: It was removed.

Point 13: L160-161 This is a consideration and should not be reported in the results section

Response: It was removed.

Point 14: TABLES 1-7 AND FIGURES 1-5. These tables and figures are structured identically and report the values of the different microbiological determinations at each sampling point. Providing a table and a figure with microbiological results for each sampling day is not very meaningful, because everything becomes very complicated and difficult to read. The authors should include all data in a maximum of one/two tables and one/two figures, so the reader can immediately visualize at a quick glance the evolution of the various microbiological parameters during the storage.

Response: In the tables is number of coliform bacteria and total count of bacteria, in the figures is number of L. monocytogenes on Oxford agar. It is different group of microorganism.

Round 2

Reviewer 1 Report

The authors have improved the manuscript considerably well and hence, accepted for publication.

Reviewer 2 Report

May be accepted in the present form.

Dear Editor

Please understand that I do not have access to professional plagiarism checking tool/software. My opinion about plagiarism is empirical only.